# Distribution and Molecular Diversity of Whitefly Species Colonizing Cassava in Kenya

**DOI:** 10.3390/insects12100875

**Published:** 2021-09-27

**Authors:** Florence M. Munguti, Dora C. Kilalo, Evans N. Nyaboga, Everlyne N. Wosula, Isaac Macharia, Agnes W. Mwango’mbe

**Affiliations:** 1Department of Plant Science and Crop Protection, University Nairobi, Kangemi, Nairobi P.O. Box 29053-00625, Kenya; dchao@uonbi.ac.ke (D.C.K.); wakesho123@gmail.com (A.W.M.); 2Kenya Plant Health Inspectorate Service, GPO, Nairobi P.O. Box 49592-00100, Kenya; macharia.isaac@kephis.org; 3Department of Biochemistry, University Nairobi, GPO, Nairobi P.O. Box 30197-00100, Kenya; nyaboga@uonbi.ac.ke; 4International Institute of Tropical Agriculture, Dar es Salaam P.O. Box 34441, Tanzania; E.Wosula@cgiar.org

**Keywords:** *Bemisia tabaci*, mtCOI gene, *P. bondari*, KASP genotyping, haplotype

## Abstract

**Simple Summary:**

The whitefly, *Bemisia tabaci* (Gennadium, Hemiptera) is a crop pest and plant-virus vector known to transmit more than 300 plant viruses. Among other plant viruses, the vector transmits viruses that cause the two major devastating viral diseases of cassava in sub-Saharan Africa namely cassava mosaic disease (CMD) and cassava brown streak disease (CBSD). In order to achieve effective implementation of whitefly management programs including prevention of spread of the species, accurate species identification is vital. Morphological identification approaches toward whitefly species limits the capacity to accurately identify new species, especially the presence of cryptic species such as the numerous *B. tabaci* genetic groups. Using the mitochondrial DNA cytochrome oxidase 1 (mtCO1) sequences, four distinct whitefly species namely *Bemisia tabaci*, *Bemisia afer*, *Aleurodicus dispersus* and *Paraleyrodes bondari* were identified in samples collected from major cassava growing regions in Kenya. The study presents the first report of *P. bondari* (Bondar’s nesting whitefly) on cassava in Kenya. We found three *B. tabaci* genetic groups of SSA1, SSA2 and Indian Ocean (IO) putative species colonizing cassava in Kenya. The information is useful to inform better management strategies of the whitefly vectors to reduce the impact of cassava viral diseases, which continue to be a threat to food security in major cassava growing regions.

**Abstract:**

The whitefly, *Bemisia tabaci* (Gennadium, Hemiptera) has been reported to transmit viruses that cause cassava mosaic disease (CMD) and cassava brown streak disease (CBSD) in many parts of sub-Saharan Africa (SSA). Currently, there is limited information on the distribution, species and haplotype composition of the whitefly populations colonizing cassava in Kenya. A study was conducted in the major cassava growing regions of Kenya to address this gap. Analyses of mitochondrial DNA cytochrome oxidase 1 (mtCO1) sequences revealed the presence of four distinct whitefly species: *Bemisia tabaci*, *Bemisia afer*, *Aleurodicus dispersus* and *Paraleyrodes bondari* in Kenya. The *B. tabaci* haplotypes were further resolved into SSA1, SSA2 and Indian Ocean (IO) putative species. The SSA1 population had three haplogroups of SSA1-SG1, SSA-SG2 and SSA1-SG3. Application of KASP genotyping grouped the *Bemisia tabaci* into two haplogroups namely sub-Saharan Africa East and Southern Africa (SSA-ESA) and sub-Saharan Africa East and Central Africa (SSA-ECA). The study presents the first report of *P. bondari* (Bondar’s nesting whitefly) on cassava in Kenya. *Bemisia tabaci* was widely distributed in all the major cassava growing regions in Kenya. The increased detection of different whitefly species on cassava and genetically diverse *B. tabaci* mitotypes indicates a significant influence on the dynamics of cassava virus epidemics in the field. The study highlights the need for continuous monitoring of invasive whitefly species population on cassava for timely application of management practices to reduce the impact of cassava viral diseases and prevent potential yield losses.

## 1. Introduction

Whiteflies (Hemiptera: Aleyrodidae) are mainly tropical insects but are also found in all warmer parts of the world, as well as in greenhouses in temperate regions [1,2]. There are more than 1500 whitefly species, of which only a few have been reported to transmit plant viruses [3,4,5]. The whitefly, *Bemisia tabaci* (Genaddius) is the most important in terms of virus transmission and has been reported to correspond to a complex of more than 40 morphologically indistinguishable species that attack several host crops [6,7,8,9,10,11,12]. This poses a challenge in developing sustainable effective control approaches since the species differ in terms of their susceptibility to natural enemies, insect resistance, fecundity and vector competence [13]. The *Bemisia tabaci* is regarded as a super vector due to the global spread of the viruses they transmit [14,15]. It is known to have over 1000 host-plant species including cassava, sweet potato, tomato, cucurbits, cotton as well as various fruits and ornamental plants [15,16]. *Bemisia tabaci* is known to transmit over 300 economically important plant viruses including *begomoviruses, criniviruses*, *carlaviruses*, *torradoviruses* and *ipomoviruses* [4,17,18,19,20].

The two major devastating viral diseases of cassava in sub-Saharan Africa, namely cassava mosaic disease (CMD) and cassava brown streak disease (CBSD), are caused by *begomoviruses* and *ipomoviruses*, respectively, and are transmitted by the whitefly, *Bemisia tabaci* [11,21]. Combined losses from the two diseases have been estimated to a value of USD 1 billion per year [22,23], threatening the livelihoods of more than one billion smallholder households that depend on cassava as a staple food crop in sub-Saharan Africa. *Bemisia tabaci* transmits cassava mosaic begomoviruses (CMBs) in a persistent manner and cassava brown streak ipomoviruses (CBSIs) in a semi-persistent manner [4,23,24,25,26,27]. The rapid spread of CMD and CBSD is associated with the super abundance of *B. tabaci* [11,28]. Other than *B. tabaci*, reports in Kenya indicate transmission of CBSI’s by other whitefly species namely spiraling whitefly (*Aleurodicus dispersus*) [26,29] and *Trialeuroides vaporariorum* [29]. The first occurrence of *Aleurodicus dispersus* in Kenya was in 2009 and was confirmed to be involved in transmission of CBSIs [26,30]. Khamisi et al. [31] identified *B. afer* and *A. dispersus* on cassava in Kenya, although their role in transmission of viruses infecting the crop has not been elucidated. In addition, a recent report of *P. bondari* on cassava in Uganda by Omongo et al. [32] indicates the need to understand its potential role in transmission of cassava viruses.

Most whitefly species studies in Africa have been mainly focused on *B. tabaci* and in sub-Saharan Africa. Two major groups of *B. tabaci* have been characterized as the “cassava type” that is unique to cassava as a host and the “non-cassava type” that colonizes other host plants such as tomato and sweet potato but does not colonize cassava [11,33,34]. The non-cassava types of *B. tabaci* that do not establish on cassava belong to groups designated as Indian Ocean (IO), Mediterranean (MED), Middle East Asia minor 1 (MEAM 1) and Uganda [11,33,35]. Although the *Bemisia tabaci* MEAM 1 and MED formerly termed biotype B and biotype Q, respectively, [17] have not been reported on cassava, they are the most widely disseminated and impactful species globally.

Five genetically distinct groups of *B. tabaci* putative species have been reported for cassava in Africa, namely sub-Saharan Africa 1 (SSA-1 to 5), SSA-2, SSA-3 SSA-4 and SSA-5) [11,12,36]. Sub-Saharan Africa 1 (SSA-1) is the most prevalent species and has been reported to occur throughout sub-Saharan Africa and has further been divided into five subgroups (SG) namely SSA1-SG1, SSA1-SG2, SSA1-SG3, SSA1-SG4 and SSA-SG5 [11,19]. Sub-Saharan Africa 2 (SSA-2) has been reported to occur in East and West Africa, SSA-3 and SSA-4 in Central and West Africa and SSA5 in South Africa [11,37,38]. In East and Central Africa, the most common cassava *B. tabaci* species have been identified to belong to the three sub-Saharan Africa taxa (SSA1, SSA2 and SSA3) [11,12,36,38] with SSA1 as the most prevalent species in the region. Despite extensive studies done on *B. tabaci* putative species in many cassava-growing countries in Eastern and Central Africa, there are limited studies on diversity of whitefly species and the circulating *B. tabaci* species haplotypes on cassava in Kenya.

To achieve effective implementation of whitefly management programs including prevention of spread of the species, accurate species identification is vital [39]. This is due to the wide variation in the species host preference, insecticide resistance profiles, pathogen vectoring range and increased fecundity of the different biotypes [13,40]. Morphological identification approaches of whitefly species are challenging and limits the capacity to accurately identify new species especially the presence of cryptic species such as the numerous *B. tabaci* mitotypes [41]. The use of mitochondrial DNA cytochrome oxidase I gene (mtCO1), first reported by Frohlich et al. [42] has been the most widely used molecular approach to distinguish different whitefly species that can be associated with specific biotypes. The region provides the highest genetic variability among the whitefly species and has been widely used in identification of the *B. tabaci* complex groups [8,11,12,32,33,41,43,44]. Use of mtCO1 gene allows *B. tabaci* sequence data to be applied to more global phylogenetic questions with a large number of mtCO1 *Bemisia* species sequences currently available in the Genbank [44]. However, mtCO1 has the limitation that it is a single locus that is maternally inherited; therefore, it is likely to yield inadequate genetic resolution to distinguish populations, and it does not provide a full delineation of phylogenetic history [45]. Therefore, there is need to complement mtCO1 markers with low cost and informative single nucleotide polymorphism (SNP)-based markers such as Kompetitive-Allele-Specific PCR (KASP) for identification and genetic diversity studies of whitefly species.

A rapid and high throughput genotyping tool based on Kompetitive-Allele-Specific PCR (KASP) approach has been reported to characterize and categorize cassava-colonizing The results were over 99% consistent with the results obtained from NextEra restriction-site associated DNA sequencing (NextRAD) [34], proving KASP genotyping assay as a reliable SNP-based assay for identification of *B. tabaci* whiteflies on cassava.

There is inadequate information on the genetic variability and geographical distribution of whitefly species colonizing cassava in different cassava growing agro-ecological zones Kenya. Data are urgently required regarding the genetic groups, haplotype diversity, geographical distribution, and the phylogenetic relationships of whitefly species in Kenya. Therefore, the aims of the study were to (i) determine the identity and distribution of the whitefly species, and (ii) the genetic diversity of *B. tabaci* complex colonizing cassava in different cassava growing regions in Kenya. This information facilitates the design of future studies that aim toward prevention and management of this economically significant pest as well as development of control strategies for the viral diseases they transmit and spread.

## 2. Materials and Methods

### 2.1. Sampling Sites, Sample Collection and Preservation

A survey was carried out between July 2018 and July 2020 to determine the whitefly species associated with cassava in different cassava growing regions in coastal (Kilifi, Kwale and Taita-Taveta counties), Eastern (Kitui, Makueni, Machakos and Embu counties), central (Murang’a county), Nyanza (Migori, Kisumu and Homa Bay counties) and Western (Busia, Bungoma and Kakamega county) parts of Kenya (Figure 1 and Appendix A). The regions sampled represented different agro-ecological zones with different rainfall and temperature regimes. Western and Nyanza region counties receive bimodal rainfall ranging from 950 to 1500 mm annually, temperatures ranging from 14.4 to 25.4 °C and altitude ranges from 100 to 1800 m above sea level. In Eastern region, altitude ranges from 100 to 1800 m with rainfall potential of 500–760 mm. The coastal region has rainfall ranging from 500 to 1000 mm annually, and a temperature range between 22.4 to 30.3 °C.

In all the sampled sites, five to ten adult whiteflies were collected from 10 randomly selected 3- to 8-month-old cassava plants per field on the underside of the uppermost five fully expanded leaves. Using a hand-held mouth aspirator (John W. Hock Company, Gainesville, FL, USA), the samples were carefully aspirated and immediately preserved in 95% ethanol in vials [30,35]. Information on the name of the County, sub-County, ward and field number together with the corresponding geographical coordinates using Global Positioning System (GPS) were recorded for each field. A total of 76 adult whitefly samples were collected with 37 from coastal region, 13 from Western, 13 from Nyanza, 9 from Eastern and 4 from the Central region (Appendix A). The number and distribution of collection sites varied depending on the number of cassava fields that were found in each region. The collected whitefly samples were transferred to Kenya Plant Health Inspectorate Service (KEPHIS)—Plant Quarantine and Biosecurity laboratory, Muguga, Kenya and kept at –20 °C awaiting molecular analysis.

### 2.2. Genomic DNA Extraction and Polymerase Chain Reaction Amplification

Genomic DNA was extracted from individual adult whiteflies using the Chelex method [46] with slight modifications. Each individual whitefly previously preserved in 95% ethanol was removed from the vial using a 200 µL pipette tip (Gilson, Middleton, WI, USA) and dried on a soft clean tissue for a few seconds. The single whitefly per sample was macerated using a clean micropestle in a microfuge with lysis buffer in 120 μL TE solution (10 mM Tris–HCl and 1 mM EDTA, pH 8.0) (FisherBiotech, Bridgewater, NJ, USA) containing 20% Chelex (BIO-RAD, Watford, UK) and 300 μg proteinase K (Jena Bioscience, Dortmund, Germany). The mixture was vortexed, and the contents spun down on a microcentrifuge (Sigma Laborzentrifugen GmbH, Osterode am Harz, Germany) and incubated at 60 °C for 15 min followed by protein denaturation at 96 °C for 10 min using Grant SUB Aqua 18 machine (Grant Instruments, Cambridge, England). The lysis mixture was centrifuged in a microcentrifuge (Sigma 1-14 model 10014; Sigma Laborzentrifugen GmbH, Osterode am Harz, Germany) at 13,000 rpm for 10 min and the lysate collected and stored at −20 °C awaiting downstream processes. The DNA quantity and quality were checked using NanodropONE (Thermo Scientific, Wilmington, DE, USA).

The PCR amplification reactions were performed using two sets of primers; the universal DNA primer pair (LCO1490: 5′-GGTCAACAAATCATAAAGATATTGG-3′ and HCO2198: 5′-TAAACTTCAGGGTGACCAAAAAATCA-3′) targeting all species [47] and the standard mtCOI primer pair (2195Bt: 5′ TGRTTTTTTGGTCATCCRGAAGT3′ and C012/Btsh2 5′-TTTACTGCACTTTCTGCC-3′) targeting mainly Bemisia species as described by Mugerwa et al. [12] amplifying ~867bp region. The PCR reaction was carried in a total of 25 μL reaction volume containing 12.5 μL OneTaq 2× MM with standard buffer (New England biolabs), 0.5 μL each of 10 μM for both forward and reverse primer and 2 μL of DNA template. This was carried out in Eppendorf master cycler machine pro 6321 (Eppendorf, Taunton, MA, USA) with thermal cycling conditions as follows: 94 °C for 5 min, 37 cycles of 94 °C for 60 sec, 53 °C for 60 sec and 72 °C for 60 sec, and final extension at 72 °C for 10 min.

The PCR amplicons were resolved on 1.5% agarose gel in 1× TBE buffer stained with 0.01% Gel Red (Biotium, Fremont, CA, USA) at 100 volts for 30 min and the amplified product gel bands visualized in a C280 gel documentation system (Azure Biosytems, Dublin, CA, USA). Amplicons of expected band sizes were estimated by comparison to a direct Load PCR 100-bp low ladder (Sigma D3687). Purification of the amplified PCR products was carried out using GeneJet Purification kit (Thermo Scientific cat K0702; Thermoscientific, Vilnius, Lithuania) according to manufacturer’s instructions. Quantification of the purified PCR product was performed using Nano drop One (Thermo Scientific, Wilmington, DE, USA) and the purified PCR products submitted for bi-directional Sanger sequencing at Macrogen (Macrogen Europe, B.V, Amsterdam, The Netherlands). All the 76 whitefly samples were successfully sequenced and used for the downstream analysis.

### 2.3. KASP Genotyping of Whitefly Samples

Thirty-eight samples representing the samples that had amplified for the different *B. tabaci* mitotypes with CO1 primers were selected and used for KASP genotyping (Appendix A). A set of six primers (BTS99-319, BTS22-762, BTS141, BTS55-473, BTS613 and BTS46203) as described by Wosula et al. [48] were used. Conventional primers were used to generate PCR products of genome portions containing target SNPs and the PCR products were then used as DNA template in KASP genotyping [48]. The KASP reaction mixture (total volume of 10 µL) contained 5 µL of 2× KASP master mix, 0.14 µL of KASP primer assay mix and 5 µL DNA template (1 µL of PCR product/DNA extract + 4 µL of sterile water).

KASP genotyping was performed in a Stratagene MX 3000P (Agilent Technologies, Santa Clara, CA, USA). The following cycling conditions were used: Stage1: 30 °C 60 sec (pre-read); Stage 2: 94 °C for 15 min hot-start Taq activation (1 cycle); Stage3: 94 °C for 20 sec, 61 °C (61 °C decreasing 0.6 °C per cycle to achieve a final annealing/extension temperature of 55 °C) for 60 sec (10 cycles); Stage 4: 94 °C for 20 sec, 55 °C for 60 sec (29 cycles); Stage 5: 94 °C for 20 sec, 57 °C for 60 sec (3 cycles); Stage 6: 37 °C for 60 sec (1 cycle, cooling) followed by an end-point fluorescent read. These conditions were used for four primers (BTS99-319, BTS22-762, BTS55-473, BTS141), while stage 3: 94 °C for 20 sec, 68 °C (68 °C decreasing 0.6 °C per cycle to achieve a final annealing/extension temperature of 62 °C) was used for two primers BTS613 and BTS46-203. The quality of genotyping cluster plots was visually assessed and only samples in distinct clusters with respective positive controls were considered for manual SNP calling using the MxPro 5 software incorporated into the Stratagene MX 3000P unit.

### 2.4. Sequence Quality Control, BLAST Search and Phylogenetic Analysis

Quality control of the resulting sequences including trimming was performed using CLC Main workbench software version 8.1.3 (QIAGEN, Aarhus, Denmark). Sequences that were not of good quality were discarded. The trimming was performed to 609 bp. The sequences were assembled and the consensus sequences resulting from the overlapping portions of high quality reads from each of the forward and reverse sequences saved as fasta format. Similarity searches were performed by querying the consensus sequences via Basic Local Alignment Search Tool (BLAST) at GenBank database hosted by the National Centre of Biotechnology Information (NCBI), Bethesda, MD, USA and on the Barcode of Life Database (BOLD) (https://www.boldsystems.org/ accessed on 20 July 2021). The BLASTn algorithm was used to identify regions of similarity between the consensus sequences from the study and the sequences available in the GenBank.

The resulting BLASTn search results were tabled with the percentage identity, E-value, percentage coverage as a well as the accession number per sample. Multiple sequence alignment was performed using Clustal W in MEGA (version 6.06) [49] for phylogenetic and molecular evolutionary analysis. Reference sequences were retrieved from the GenBank and included in the analysis [50]. The maximum likelihood algorithm based on Kimura 2-parameter model [51] was used to infer the evolutionary relationship between the various samples and phylogenetic tree constructed using 1000 bootstrap replicates in MEGA (version 6.06). The representative resulting sequences were submitted to GenBank through Bankit in the online submission portal [52]. The extent of nucleotide sequence variation within the identified *B. tabaci* (groups with at least 5 sequences) and *B. afer* was examined. Estimates were obtained for the number of haplotypes, polymorphic sites (S), average number of nucleotide differences (k), nucleotide diversity (Pi), haplotype diversity (Hd), Theta per sequence and Theta per site. Tajima’s D and Fu’s Fs were obtained using the mismatch distribution procedure of Dna-SP 6.12.03 [53].

## 3. Results

### 3.1. Presense of Whiteflies in Inspected Cassava Fields

Samples of adult whiteflies were collected in all the inspected cassava growing regions in Kenya (Figure 1 and Appendix A). Adult whiteflies were predominantly observed on the underside of the young upper cassava leaves (Figure 2A,B). For the spiraling whiteflies, characteristic spiraling pattern of the eggs spirals was evident on the underside of the leaves (Figure 2C,D). Characteristic wilting and sooty mold caused by heavy whitefly infestation was observed on cassava leaves and stems (Figure 2E,F).

### 3.2. Determination of Whitefly Species and Phylogenetic Analysis

Amplification of partial mtCO1 gene for each of the adult whitefly using primer pair LCO1490/HCO2198 [48,54] gave the expected size of the amplicons of 710 bp. A total of 75 mtCO1 high-quality sequences from individual whiteflies were used in the subsequent analysis. Similarity search of the 75 new mtCO1 generated in the current study, through BLASTn in NCBI and in the BOLD systems database revealed four different whitefly species (*B. tabaci*, *B. afer*, *P. bondari* and *A. dispersus*) with percentage similarities between 99–100%. Two of the whitefly species (*A. dispersus* and *P. bondari*) identified were non-*Bemisia*. Phylogenetic analysis conducted on mtCOI sequences of the non-*Bemisia* whitefly species with already defined sequences from the GenBank, grouped our non-*Bemisia* whitefly species into two clusters based on the topology of the phylogenetic tree (Figure 3). Representative sequences for *A. dispersus* and *P. bondari* were deposited in the Genbank with accessions MZ331821-MZ331823 and MZ 329 998-MZ330000, respectively.

Alignment of sequences generated using primers specific for *Bemisia* followed by phylogenetic analysis revealed three genetic groups of *B. tabaci* complex, which included: sub-Saharan Africa 1 (SSA1), sub-Saharan Africa 2 (SSA2) and Indian Ocean (IO) mitotypes (Figure 4). The members of SSA1 *B. tabaci* grouped into three clusters/subgroups of SSA1-SG1, SSA1-SG2 and SSA1-SG3 (Figure 4). Another cassava whitefly species identified was *Bemisia afer*, which clustered distantly from the SSA1, SSA2 and Indian Ocean subgroups (Figure 4).

Using KASP genotyping, the tested whitefly samples grouped into two clusters namely SSA-ESA and SSA-ECA whereby SSA-ECA composed of the subgroups SSA1-SG1 and SSA1-SG2 whereas SSA-ESA comprised of SSA1-SG3 (Appendix A). The mitotype SSA2 was not tested using KASP as samples failed to amplify due to degradation of DNA during transportation for KASP genotyping.

### 3.3. Distribution of Whitefly Species and Bemisia tabaci Haplotypes

Of the four whitefly species identified based on mtCO1 sequences, *B. tabaci* the most abundant, found in 70% of the sequenced samples followed by *B. afer* (13%), *A. dispersus* (8%) and *P. bondari* (5%). *Bemisia tabaci* was widely distributed in all the counties of the surveyed major cassava growing regions in Kenya (Figure 5A). *Bemisia afer* was identified in Nyanza region (Migori and Homabay counties) and in Eastern (Makueni and Machakos counties) and Central (Murang’a county). The two non-*Bemisia* type of whiteflies (*A. dispersus* and *P. bondari*) identified in the study were only found in samples collected from the coastal region of Kenya. The spiraling whitefly (*A. dispersus)* was the most predominant non-Bemisia whitefly species and widely distributed in all the three counties (Kwale, Kilifi and Taita-Taveta) of coastal Kenya (Figure 5A). The *P. bondari* was detected in only three samples from Taita Taveta and Kilifi counties.

The *B. tabaci* haplotype distribution per region is shown in Figure 5B. The major *B. tabaci* mitotype cluster was SSA1-SG3, which was predominant in coastal and eastern regions of Kenya with 90% and 50% samples, respectively, clustering under this subgroup. The subgroup SSA1-SG3 was not identified from any whitefly samples collected in western and Nyanza region of Kenya. SSA1-SG2 was identified on whitefly samples from one county each in western (Busia county) and Nyanza (Migori county). SSA1-SG1 was predominant in Western and Nyanza region of Kenya especially in the counties of Busia, Kakamega, Migori, Kisumu and Homabay. Only two whitefly samples were identified as SSA-SG1 in Coastal region of Kenya (one from Taita-Taveta and the other from Kwale County). The subgroup SSA1-SG1 was not identified from any tested whitefly samples from Eastern and Central region of Kenya (Figure 5B). The sub-Saharan Africa subgroup 2 (SSA2) was identified in six whitefly samples from Western and Nyanza region of Kenya (Busia, Bungoma, Homabay and Kakamega counties), and in one sample collected in the coastal region (Taita-Taveta county) of Kenya. SSA2 was not detected in any of the samples from Eastern and Central region of Kenya. The Indian Ocean (IO) putative species was detected only in two whitefly samples from Eastern region of Kenya.

### 3.4. Population Genetic Analysis

Population genetics analysis of whitefly species *B. tabaci* and *B. afer* revealed six *B. tabaci* mtCO1 haplotypes, although only SSA1-SG1 and SSA1-SG3 haplogroups had more than four sequences. SSA1-SG1 had only one haplotype with 17 samples, while SSA1-SG3 had two haplotypes with 25 and 6 samples, respectively. The Tajima’ D *p*-value was significant for all *B. tabaci* indicating no deviation from the expectation of neutral selection, but values were non-significant for SSA1-SG3 indicating no deviation from the expectation of neutral selection. *Bemisia afer* had only two haplotypes each with 8 and 2 samples, respectively and the Tajima’ D *p*-value was non-significant indicating no deviation from the expectation of neutral selection (Table 1).

## 4. Discussion

This study is the first extensive sampling and molecular identification of whitefly species colonizing cassava along a geographical transect from coastal Kenya through eastern, central, Nyanza to western part of Kenya. Using partial mtCO1 gene sequences [48], our study revealed the occurrence of four different species of whitefly (*Bemisia tabaci*, *Aleurodicus dispersus*, *Bemisia afer* and *Paraleyrodes bondari*) colonizing cassava in Kenya. The whitefly species, *B. tabaci* was the most abundant whitefly species colonizing cassava, similar to previous reports by Mware et al. [26] and Njoroge et al. [27]. The study by Njoroge et al. [27], identified *B. tabaci* and *Trialeurodicus vaporarium* in whitefly samples collected from different agro-ecological zones in cassava growing regions of Kenya. However, the whitefly species *Trialeurodicus vaporarium* was not detected in any of the whitefly samples collected in our study. Recent findings by Khamis et al. [31] reported *A. dispersus* and *B. afer* on whitefly samples collected on cassava host in Kenya, and did not detect any *B. tabaci*, probably due to the limited number of whitefly samples collected on cassava host. The genetic diversity and geographical distribution of *B. tabaci* are of great interest because of its importance as insect vector for cassava brown streak ipomoviruses and cassava mosaic begomoviruses [11,23]. The use of mtCO1 primers specific to *Bemisia tabaci* [12] in this study, revealed three genetic groups of *B. tabaci* haplotypes (SSA1, SSA2 and IO). The species SSA1 was the most abundant *B. tabaci* haplotype with three subgroups (SSA1-SG1, SSA-SG2 and SSA1-SG3) identified from the whitefly samples. Previous studies in Kenya by Manani et al. [55], confirmed *Bemisia tabaci* species belonging to five distinct clades within the sub-Saharan Africa (SS1 and SS2) genetic groups, but could not link the different clades to the specific SSA1 haplotypes. SSA1 and SSA2 *Bemisia tabaci* haplotypes have been reported to colonize cassava and are vectors of viruses associated with CMD and CBSD [24].

Among the SSA-1 haplotypes of *B. tabaci* reported in this study, SSA1-SG3 sub-group was the predominant and was found in samples collected from Coastal and Eastern regions of Kenya. SSA1-SG3 has been reported to be the indigenous *B. tabaci* species in coastal regions of the East Africa [21,56]. SSA1-SG1 sub-group was the second most frequent SSA1 *B. tabaci* haplotype and was found mostly in samples from Western and Nyanza region. The subgroup SSA1-SG2 was identified only in three samples from Western region with none identified in Eastern, central and coastal region of Kenya.

The sub-Saharan Africa 2 (SSA2) genetic group was found in whitefly samples collected from Western region of Kenya and only one sample collected from Coastal region of Kenya. This is contrary to previous studies that reported SSA2 as the most diverse haplotype in geographical coverage in Eastern and Central Africa regions [55,57]. The presence of SSA2 haplotype in coastal region of Kenya for the first time could be associated with the possibility of inadvertent movement whitefly nymphs or eggs on leaf material transported with cassava stems during the exchange of cassava planting material by farmers. Previous reports in Uganda have associated SSA2 genetic group with regions having high incidences and severity of cassava mosaic disease [58]. Future studies need to be conducted to explore the role SSA2 genetic group in single and/or dual transmission of viruses causing CMD and CBSD, in order to inform management strategies for the diseases they spread.

Various non-cassava *B. tabaci* haplotypes have been reported on cassava and other hosts including Mediterranean (MED) formerly biotype Q, Middle-East Asia Minor 1 (MEAM-1), Indian Ocean (IO) and Uganda [33,35]. Our study identified the Indian Ocean (IO) putative species from only two samples collected in Eastern and Central region of Kenya. Homology search on our sequences BLASTn search and BOLD systems shared similarity percentages of 100%, with *B. tabaci* Indian Ocean on tomato (Accession Number AJ550171.1). The Indian Ocean whitefly species has been reported as a non-cassava colonizer and is mostly found on sweet potato and other vegetables [59]. The Indian Ocean (IO) species are unable to reproduce on and colonize or utilize cassava and therefore their presence on cassava in this study could be due to the fact that they are transient individuals moving through cassava in search of the preferred hosts such as tomato and sweet potato.

Using KASP genotyping, the representative samples of *B. tabaci* haplotypes grouped into two clusters; sub-Saharan Africa, East and central Africa (SSA-ECA) and sub-Saharan Africa, East and Southern Africa (SSA-ECA) whereby SSA-ECA composed of the subgroups SSA1-SG1 and SSA1-SG2 whereas SSA-ESA comprised of only SSA1-SG3. The results were consistence with previous studies [46,57]. SSA-ECA comprising of SSA1-SG1 and SSA1-SG2 have been reported in areas with severe CMD and CBSD [46]. SSA1 –SGA and SSA1-SG2 (SSA-ECA) in western region of Kenya indicates a possible correlation with the high incidence and severity of cassava viral diseases in the region [60,61].

The wide distribution of *B. tabaci* in all major cassava growing regions of Kenya, suggest the ability of this species to adapt to many different agro-ecological zones, contrary to other species detected in this study (*B. afer*, *Aleurodicus disperses* and *Paraleyrodes bondari*). From this study, *Bemisia afer* was reported as the second most abundant whitefly species and was more frequent on samples collected from Eastern and Western region with none identified from samples collected in the coastal region of Kenya. Previous studies have associated *B. afer* with cassava as a host [27,31,62] as well as insect vector in transmission of Sweet Potato Chlorotic Stunt Virus (SPCSV) [63]. There is therefore a need for transmission studies to explain the possible role of *B. afer* whitefly species in transmission of viruses associated with CMD and CBSD. Spiraling whitefly *Aleurodicus dispersus* Russel (Homoptera, Aleyrodidae) identified in this study was predominantly from samples collected in the coastal region of Kenya. *Aleurodicus dispersus* has been associated with the resurgence of cassava brown streak disease and its role as a vector for the cassava brown streak viruses in the coastal region of Kenya was confirmed by Mware et al. [26].

This study identified *Paraleyrodes bondari* (Bondar’s nesting whitefly) in four whitefly samples collected from the coastal regions. Although we did not analyze nymph samples, the failure to detect this species in other regions suggests a restricted occurrence. Homology search on our sequences BLASTn search and BOLD systems shared similarity percentages of 99.84% and 100%, respectively, with *P. bondari* isolates from India on guava (Accession Number MW488201.1) and coconut (Accession number MW488198). Based on phylogenetic analysis, the identified *P. bondari* isolate was closely related to *P. bondari* and *P. minei* isolates but distantly related to *A. dispersus* whitefly species based on the clustering on the phylogeny tree. This is in agreement with previous reports, confirming the identity of the isolates detected in this study [32,64,65]. This is the first report of Bondar’s nesting whitefly species, *Paraleyrodes bondari* (Hemiptera Aleyrodidae) colonizing cassava in Kenya, specifically in the coastal region of Kenya.

Bondar’s nesting whitefly, *Paraleyrodes bondari* has in previous studies been reported on coconut and guava hosts, respectively [64,66]. These plant species (coconut and guava) are important crops commonly grown by majority of farmers in the coastal region of Kenya where *P. bondari* whitefly species was identified and could have been the possible main host for the whitefly species. Host adaptation may be a more important component affecting the low predominance of *P. bondari* in cassava. In coastal Kenya, cassava is predominantly grown as a subsistence crop, usually side by side with coconut, guava and other vegetable crops. The role of other host plants was a factor that we did not investigate—*P. bondari* adults may have moved from sources other than the identified source (cassava). For example, coconut and guava hosts may sustain reproduction and development of *P. bondari* whitefly species, with adults migrating/dispersing to cassava and therefore the *P. bondari* whitefly occurring on cassava were present as visitors and were not colonizing the crop. However, recent reports on occurrence and distribution of *P. bondari* on cassava in Uganda by Omongo et al. [32] and Isabirye [67] indicates a possibility of adaptability of the whitefly species on cassava crop. Being an invasive whitefly species, further extensive surveillance to map the species distribution and possibly elucidate the possible origin, host range and the possible role in transmission of cassava viruses or viruses infecting other crops in Kenya.

Even though we detected *Paraleyrodes bondari* associated to cassava, we cannot infer its potential to effectively colonize this host since only adults were collected. In addition, the very low frequency with which it was detected, indicate that *Paraleyrodes bondari* is poorly adapted to cassava. It will be important to monitor *Paraleyrodes bondari* populations in cassava over the next years, to assess its possible adaptation to this host. Further sampling in those sites or free-choice experiments is necessary to confirm the potential of *Paraleyrodes bondari* to colonize and utilize cassava.

Previous studies have reported occurrence of different whitefly species on cassava crop [26,28,31,32,68,69,70]. Transmission of cassava viruses by some of the whitefly species has also been confirmed [24,26]. There is therefore need for studies to infer the possible role of the reported whitefly species in transmission of cassava viruses and especially the cassava brown streak ipomoviruses causing cassava brown streak disease, which has continued to be a threat to many cassava-growing regions. Based on the results from this study, there is therefore need for further surveys to identify the whitefly species on cultivated and non-cultivated hosts, as this could have implications on the dynamics of the emergence, re-emergence and spread of cassava viruses—especially the cassava brown streak ipomoviruses causing brown streak disease in cassava.

## 5. Conclusions

In this study, four different whitefly species *B. tabaci*, *B. afer*, *Aleurodicus dispersus* and *P. bondari* were identified on the whitefly samples collected from cassava in Kenya. This is the first report of *P. bondari* (Bondar’s nesting whitefly) on cassava in Kenya. Even though we detected *Paraleyrodes bondari* associated to cassava, we cannot infer its potential to effectively colonize this host since only adults were collected. *Bemisia tabaci* species complex was the most prevalent and widely distributed in all the regions (Coastal, Western, Nyanza, Eastern and Central). Three genetic groups of *B. tabaci*: SSA1 (SSA1-SG1, SSA1-SG2 and SSA1-SG3), SSA2 and Indian Ocean putative species were identified. The major *B. tabaci* haplotype was SSA1-SG3, which was predominant in coastal and Eastern regions. Our study also demonstrated application of KASP genotyping as a rapid tool for analysis of cassava *B. tabaci* in laboratories with limited access to sequencing technologies. The information will be useful to inform better management strategies of the vectors to reduce the impact of the viral diseases, which continue to be a threat to food security in major cassava growing regions.

## Figures and Tables

**Figure 1 insects-12-00875-f001:**
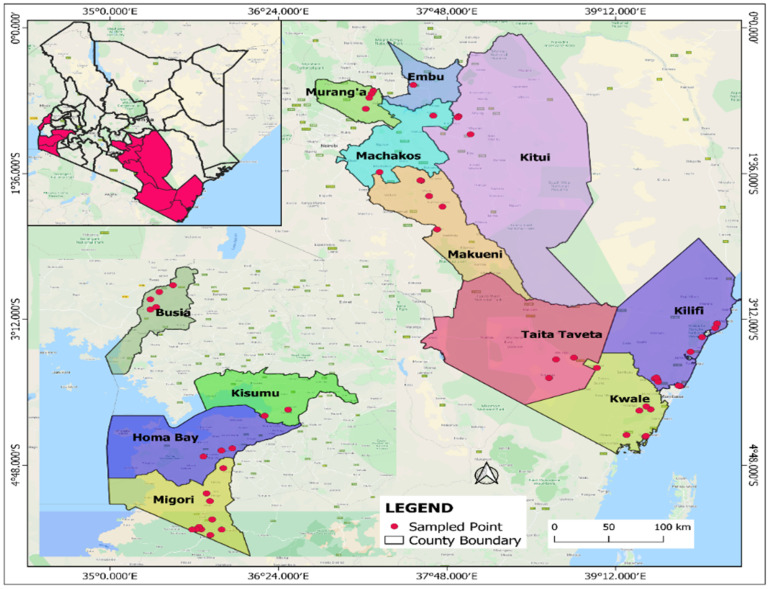
A map of Kenya showing the locations of sampling points for the whitefly species in the study.

**Figure 2 insects-12-00875-f002:**
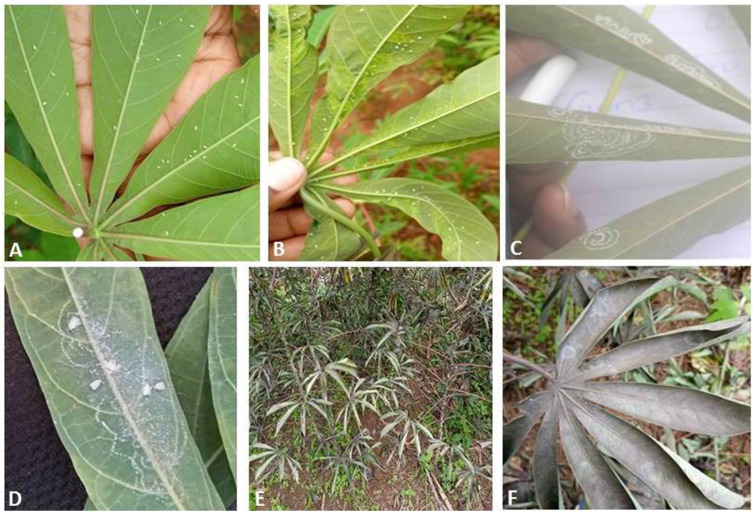
Adult whiteflies on the underside of cassava leaves (**A**,**B**). Spiraling marks left by spiraling whitefly species on the underside of cassava leaves (**C**,**D**). Characteristic wilting and sooty mold fungus caused by heavy whitefly infestation on cassava plants (**E**,**F**).

**Figure 3 insects-12-00875-f003:**
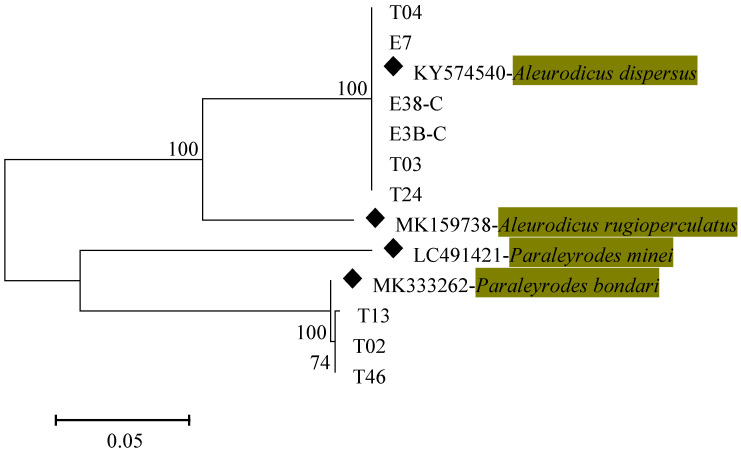
Maximum likelihood phylogenetic tree constructed using mitochondrial cytochrome oxidase 1 sequences obtained from non-*Bemisia* whitefly species collected on cassava plants in Kenya between July 2018 and July 2020. Reference sequences (◆) from NCBI were included for comparison.

**Figure 4 insects-12-00875-f004:**
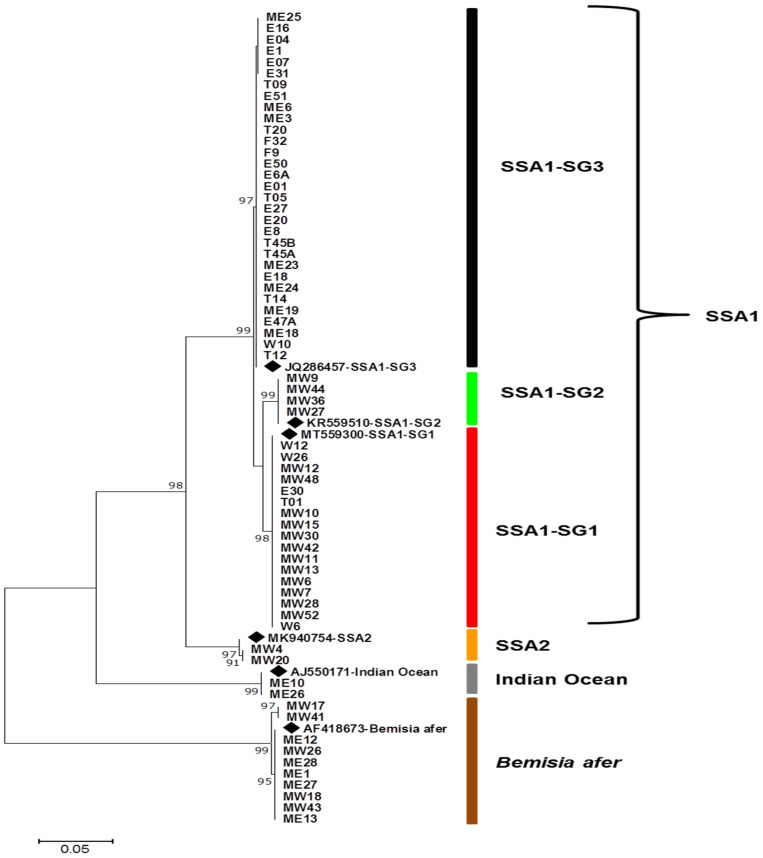
Maximum likelihood phylogenetic tree constructed using mitochondrial cytochrome oxidase 1 sequences obtained from *Bemisia tabaci* (cassava types) collected in this study. Reference sequences (◆) from NCBI were included for comparison. Only selected sequences are shown in the phylogenetic tree.

**Figure 5 insects-12-00875-f005:**
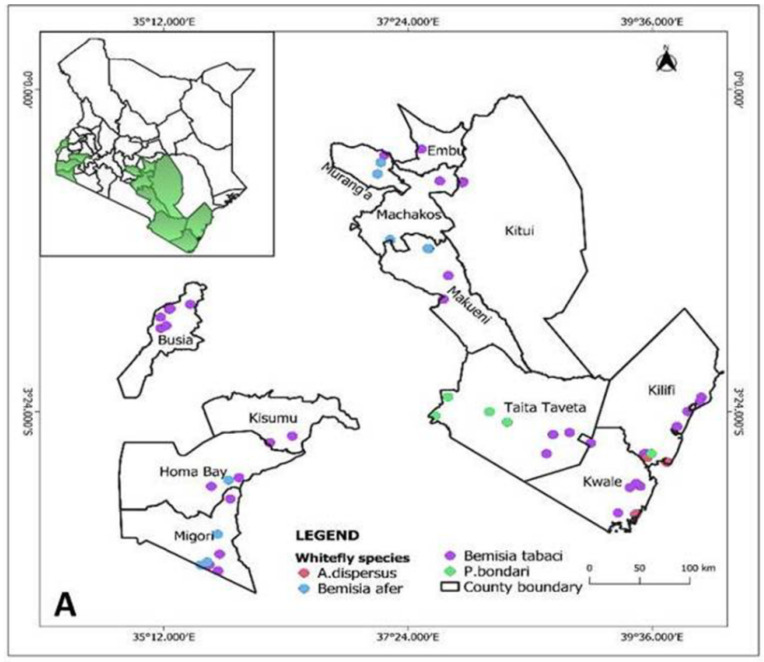
A map of Kenya and cut-out maps of cassava growing counties in Kenya showing distribution of the four whitefly species (**A**) and *B. tabaci* haplotypes (**B**) of cassava-colonizing whitefly identified from the samples collected during the study.

**Table 1 insects-12-00875-t001:** Population genetic analysis of whitefly species *Bemisia tabaci* and *B. afer* found colonizing cassava in Kenya.

Parameter	All *B. tabaci*	SSA1-SG1	SSA1-SG3	*B. afer*
Sample size	56	17	31	10
Number of haplotypes	6	1	2	2
Polymorphic sites (S)	162	0	1	5
Average no of nucleotide differences (k)	19.24091	0.0000	0.3226	1.77778
Nucleotide diversity (Pi)	0.0245	0.0000	0.0004	0.0023
Haplotype diversity (Hd)	0.702	0	0.323	0.356
Variance of Hd	0.0017	0.000	0.0078	0.0253
Standard deviation of Hd	0.0079	0.000	0.088	0.159
Theta per sequence	39.1849	-	0.2503	1.7674
Theta per site	0.0499	-	0.0003	0.0023
Fu’s Fs statistic	25.868	-	0.864	3.636
Tajima’s D	−1.80960	-	0.44525	0.02348
*p*	*p* < 0.05	-	*p* > 0.10	*p* > 0.10

## Data Availability

Partial MtCOI Sequences of the whitefly species including the *B. tabaci* haplotypes obtained in this study were deposited in Genebank through online submission portal under accession numbers (MZ33092-096, MZ331821-823, MZ331 385-396, and MZ329998-3000).

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
