# Peer review of "Distribution and Molecular Diversity of Whitefly Species Colonizing Cassava in Kenya"

_insects, 2021, doi:10.3390/insects12100875_

Round 1

Reviewer 1 Report

The paper, “Distribution and molecular diversity of whitefly species colonizing cassava in Kenya and application of KASP genotyping to identify Bemisia tabaci haplotypes,” describes a study filling a couple of useful purposes. The first is the characterization of whitefly species and sub-populations on cassava within Kenya, and the second is the demonstration of how Kompetitive allele-specific PCR (KASP) can be used as a less resource-intensive tool for characterizing population genetics. The characterization of whiteflies is especially useful for the management of whitefly-transmitted viruses in cassava, including cassava mosaic disease and cassava brown streak disease, as the identification of genetic groups associated with greater disease risk and/or severity can be used to focus management efforts to specific localities. The paper is well written, and I have some minor suggestions for revision.

The introduction sets up CO1 and KASP can be used to characterize genetic populations but doesn’t set up the premise for why you chose to use both in your study. It could be helpful to address why you chose to use both: will KASP add something to the traditional CO1 approach like greater resolution and ability to distinguish sub-populations or greater accuracy?

Within the Materials and Methods section, I wondered why were so few whiteflies collected from the Eastern and Central regions? I might suggest addressing that briefly. I also thought it would be helpful to include a supplemental table listing total samples collected per region, as well as how many per region were successfully sequenced and used in downstream analyses, and how many per region were used for KASP genotyping. Additionally, how were the 38 samples which were used in KASP genotyping chosen?

In Table 1, were the Fu’s Fs statistics listed significant? It would be helpful to see these p-values. Additionally, do the high Fu’s Fs and negative Tajima’s D values potentially indicate population expansion after a recent bottleneck, a selective sweep, or something to that extent?

Lastly, with regards to finding the B. tabaci IO species on cassava, do the authors think this could be evidence of previously unobserved cassava host-plant utilization or were the samples more likely transient individuals moving through cassava in search of preferred hosts?

Additional suggestions:

Line 12: please italicize Bemisia tabaci.

Line 23: please change Bondari to bondari.

Line 49: please insert “are” to read “…are mainly tropical insects, but are also found in all…”

Lines 57-58: please change to “The B. tabaci complex is regarded as a super vector due to the global spread of the viruses it can transmit [14,15].”

Lines 58-59: please change to “It is known to have over 1000 host-plant species…”

Lines 73,76: please write out the first occurrences of Aleurodicus and Paraleyrodes in the Introduction section.

Line 80: please change “widely experienced variation” to “wide variation”.

Lines 92-93: please change the first sentence into two separate sentences, so that the second sentence begins, “Two major groups of B. tabaci have been characterized…”

Line 99: please insert “the” to read “…they are the most widely disseminated and impactful species globally.”

Line 114: please capitalize the s in Specific.

Line 128: please change carried to carried out.

Line 144: please capitalize the p in Positioning.

Line 180: please change suing to using.

Line 218: please change Barcorde to Barcode.

Line 223: please insert the “)” for (version 6.06; [51]).

Lines 309-324: would be helpful to indicate here that Fig. 6B shows the location of subgroups by region.

Line 334: please change whiefly to whitefly.

Line 379: please change “contrary with” to “contrary to”.

Lines 405-406: please change “on the contrary to” to “in contrast to” or “contrary to”.

Line 412: please insert “of” to read “the possible role of B. afer”.

Author Response

24th August, 2021

We appreciate the time and effort by the reviewer for reviewing this manuscript and for providing insightful comments and suggestions to improve its quality. We have revised the manuscript as per the suggestions and comments. The point-by-point responses are provided below.

Comments and Suggestions for Authors

General comment on the paper: The paper, “Distribution and molecular diversity of whitefly species colonizing cassava in Kenya and application of KASP genotyping to identify Bemisia tabaci haplotypes,” describes a study filling a couple of useful purposes. The first is the characterization of whitefly species and sub-populations on cassava within Kenya, and the second is the demonstration of how Kompetitive allele-specific PCR (KASP) can be used as a less resource-intensive tool for characterizing population genetics. The characterization of whiteflies is especially useful for the management of whitefly-transmitted viruses in cassava, including cassava mosaic disease and cassava brown streak disease, as the identification of genetic groups associated with greater disease risk and/or severity can be used to focus management efforts to specific localities. The paper is well written, and I have some minor suggestions for revision.

Response: Thank you for your positive and encouraging general comments. We have incorporated the following specific comments in the revised version of manuscript.

Specific comments

  1. The introduction sets up CO1 and KASP can be used to characterize genetic populations but doesn’t set up the premise for why you chose to use both in your study. It could be helpful to address why you chose to use both:

Response: The use of mitochondrial DNA cytochrome oxidase I gene (mtCO1), first reported by Frohlich et al. (1999) has been the most widely used molecular approach to distinguish different whitefly species that can be associated with specific biotypes. The region provides the highest genetic variability among the whitefly species and has been widely used in identification of the B. tabaci complex groups. However, mtCO1 has the limitation that it is a single locus that is maternally inherited; therefore, it is likely to yield inadequate genetic resolution to distinguish populations, and it does not provide a full delineation of phylogenetic history (Ballard and Whitlock, 2004). Therefore, there is need to complement mtCO1 markers with affordable and informative/more robust markers for genetic diversity studies of whitefly species. The use of both mtCO1 and KASP in this study was to highlight the need to switch to more robust techniques such as KASP for B. tabaci. KASP is a SNP-based novel and rapid in house technique that distinguishes these whiteflies more accurately as opposed to mtCO1 as explained in Wosula et al. (2020).

will KASP add something to the traditional CO1 approach like greater resolution and ability to distinguish sub-populations or greater accuracy?

Response: KASP is a SNP based technique that is highly sensitive compared to other techniques. It was developed using sequence data generated through SNP genotyping (NextRAD) to characterise cassava colonizing Bemisia tabaci. SNP genotyping which is more robust compared to mtCO1 sequencing revealed six major haplogroups of cassava whiteflies. The mtCO1 was ineffective at distinguishing some of the haplogroups for example mitotype SSA-SG1 was found to have whiteflies that grouped into three distinct halopgroups SSA-ECA, SSA-CA and SSA-WA. In this study mitotypes SSA1-SG1 and SSA1-SG2 are all haplogroup SSA-ECA consistent to what has been reported in Wosula et al 2017, Chen et al 2019, Wosula et al 2020. KASP was developed to distinguish these SNP-based haplogroups under standard laboratory settings, and with end point fluorescent reads eliminates the cost and time delay incurred with mtCO1 sequencing. In this study both mtCOI and KASP were used to highlight the use of KASP as a robust novel and rapid tool for identification of cassava B. tabaci.

KASP is a technique that was developed by LGC Genomics Teddington-UK, and it is based on allele-specific oligo extension and fluorescent resonance energy transfer (FRET) for signal generation. The assay uses allele-specific fluorescent labelled primers that allow for end-point fluorescent reads enabling the bi-allelic scoring of single nucleotide polymorphisms at specific loci (https://biosearch-cdn.azureedge.net/assetsv6/KASP-genotyping-chemistry-User-guide.pdf). KASP assays are low cost, high-throughput and have high specificity and sensitivity relative to other markers (Semagn et al., 2013).

  1. Within the Materials and Methods section, I wondered why were so few whiteflies collected from the Eastern and Central regions. I might suggest addressing that briefly.

Response: Few whitefly samples were collected from eastern and central regions because these are not the major cassava growing regions in Kenya and cassava fields in these regions were few. The number and distribution of collection sites varied depending on the number of cassava fields that were found in each region.

I also thought it would be helpful to include a supplemental table listing total samples collected per region, as well as how many per region were successfully sequenced and used in downstream analyses, and how many per region were used for KASP genotyping.

Response: Thank you. We have modified supplementary Table S1 and included total samples collected and sequenced collected per region. Supplementary Table S2 also been revised to reflect the number of samples per region were used for KASP genotyping.

Additionally, how were the 38 samples which were used in KASP genotyping chosen?

Response: Samples used for KASP genotyping were selected based on mtCO1 sequences. Based on mtCO1 sequences, representatives of similar mtCO1 sequences were selected for each of the SS1 haplotypes (SSA1-SG1, SSA1-SG2, SSA1-SG3). In this study, KASP genotyping results indicated that mitotypes SSA1-SG1 and SSA1-SG2 are all haplogroup SSA-ECA consistent to Wosula et al. (2017, 2020) and Chen et al. (2019).

  1. In Table 1, were the Fu’s Fs statistics listed significant? It would be helpful to see these p-values.

Response: DnaSP does not generate any p values for Fu’s Fs, this was confirmed by checking our previous analyses.

Additionally, do the high Fu’s Fs and negative Tajima’s D values potentially indicate population expansion after a recent bottleneck, a selective sweep, or something to that extent?

Response: DnaSP did not generate any p values so we cannot make any further interpretation on the Fu’s Fs statistics.

  1. Lastly, with regards to finding the tabaci IO species on cassava, do the authors think this could be evidence of previously unobserved cassava host-plant utilization or were the samples more likely transient individuals moving through cassava in search of preferred hosts?

Response: Thank you for the observation. The Indian Ocean (IO) species do not colonize or utilize cassava and therefore their presence on cassava in this study could be due to the fact that they are transient individuals moving through cassava in search of the preferred hosts. Homology search on our sequences BLASTn search and BOLD systems shared similarity percentages of 100%, with B. tabaci Indian Ocean on tomato (Accession Number AJ550171.1).

Additional suggestions:

Line 12: please italicize Bemisia tabaci.

Response: We have italicized “Bemisia tabaci’ (Line 13)

Line 23: please change Bondari to bondari.

Response: We have changed “Bondari” to “bondari” (Line 23)

Line 49: please insert “are” to read “…are mainly tropical insects, but are also found in all…”

Response: We have inserted “are” to read …are mainly tropical insects, but are also found in all…” (Line 49).

Lines 57-58: please change to “The B. tabaci complex is regarded as a super vector due to the global spread of the viruses it can transmit [14,15].”

Response: We have inserted “The” as per your comment   on page 2 of 16 of the revised manuscript, line 57.

Lines 58-59: please change to “It is known to have over 1000 host-plant species…”

Response: We have modified the sentence in line 58 of the revised manuscript from [“It is known to have over a host of over 1000 plant species”] to read… [“It is known to have over 1000 host-plant species”].

Lines 73, 76: please write out the first occurrences of Aleurodicus and Paraleyrodes in the Introduction section.

Response: Thank you. We have included the first occurrences of Aleurodicus disperses in Kenya. The first occurrence of Aleurodicus dispersus in Kenya was in 2009 and was confirmed to be involved in transmission of CBSIs (Mware et at., 2009, 2010). However, there is no previous report on the occurrence of Paraleyrodes bondari in Kenya except in the neighouring country, Uganda. This study is the first to report the occurrence of Paraleyrodes bondari in Kenya.

Line 80: please change “widely experienced variation” to “wide variation”.

Response:  We have changed “widely experienced variation” to “wide variation” (Lines 80 – 81).

Lines 92-93: please change the first sentence into two separate sentences, so that the second sentence begins, “Two major groups of B. tabaci have been characterized…”

Response: We have spit the first sentence into two separate sentences (Line 92 – 95). The second part of the sentence begins, “Two major groups of B. tabaci have been characterized as “cassava type” that is unique to cassava as a host and “non-cassava type” that colonize other host plants such as tomato and sweetpotato but do not colonize cassava” [11, 41, 42].

Line 99: please insert “the” to read “…they are the most widely disseminated and impactful species globally.”

Response: We have inserted “the” (Line 99).

Line 114: please capitalize the “s” in Specific.

Response: We have capitalized “s” in “Specific” (Line 114) of the revised manuscript. It now reads…… “Kompetitive-Allele-Specific PCR (KASP)”

Line 128: please change carried to carried out.

Response:  We have changed [“carried”] to [“carried out”] (Line 128). “….A survey was carried out between July 2018 and July 2020 to determine the whitefly species associated………”.

Line 144: please capitalize the p in Positioning.

Response: We have capitalized “p” in “Positioning” (Line 144), “Global Positioning System (GPS)”.

Line 180: please change suing to using.

Response: We apologize for the typographical error. We have changed “suing” to “using” (Line 180).

Line 218: please change Barcorde to Barcode.

Response: We have corrected the error from “Barcorde” to “Barcode” as per your comment. (Please see page 5 of 16, in line 218 of the revised manuscript)

Line 223: please insert the “)” for (version 6.06; [51]).

Response: This is noted and we have inserted “ (Line 223).

Lines 309-324: would be helpful to indicate here that Fig. 6B shows the location of subgroups by region.

Response: Thank you for the observation. The figure has been revised to Fig. 5B.  We have revised lines 309 – 324 and indicated that Fig. 5B shows the location of the B. tabaci haplotypes per region.

Line 334: please change whiefly to whitefly.

Response: We have noted this anomaly and we have changed “whiefly” to “whitefly” (Line 344).

Line 379: please change “contrary with” to “contrary to”.

Response:  We have changed “contrary with” to “contrary to” (Line 329).  “This is contrary to previous studies that reported SSA2 as the most diverse haplotype in geographical coverage in Eastern and Central Africa regions [57, 59].

Lines 405-406: please change “on the contrary to” to “in contrast to” or “contrary to”.

Response: We have changed “on the contrary to” to “contrary to” (Lines 405-406). The wide distribution of B. tabaci in all major cassava growing regions of Kenya, suggest the ability of this species to adapt to many different agro-ecological zones contrary to other species detected in this study (B. afer, Aleurodicus disperses and Paraleyrodes bondari).

Line 412: please insert “of” to read “the possible role of B. afer”.

Response: We have noted the anomaly and inserted “of” (Line 412). There is therefore a need for transmission studies to explain the possible role of B. afer whitefly species in transmission of viruses associated with CMD and CBSD”.

Reviewer 2 Report

In the manuscript “Distribution and molecular diversity of whitefly species colonizing cassava in Kenya and application of KASP genotyping to identify Bemisia tabaci haplotypes” the authors collect the samples of whiteflies across Kenya within several years and analyse the mitochondrial haplotypes. Due to challenging morphological species identification in whitefly, molecular methods are essential for this purpose. The authors carry out solid basic research in population genetics which is absolutely necessary for further pest management. The manuscript is well-written, easy to read and I suggest that is suitable for publication in the Insects journal. However, I have some suggestions and comments:

I suggest shortening the title to its first part: “Distribution and molecular diversity of whitefly species colonizing cassava in Kenya” as KASP is used here as not the only method, it is not used for the first time and there is another publication by Wosula et al. 2020 which is dedicated to the application of KASP genotyping in Bemisia tabaci.

For the collected samples, it would be good if the authors make an additional table in the supplementary to list all the samples with the locations, years, species, sample size, coordinates and other relevant information.

I understand the information may be scarce but it would be good to add more discussion on species yearly distribution and population dynamics (if there is any knowledge on it) as you use samples of several years and they clearly differ across different years.

The authors could add a couple of sentences why they chose KASP genotyping over the other available methods

Page 102 - The hyplotype names need the same punctuation: (SSA-1 to 5), SSA-2, SSA-3, SSA-4 and SSA-5.

Page 144 - Global Positioning System - all capitals

Figure2 – place pictures in alphabetical order

Figure3 is missing

Figure4 – make the same font

Author Response

24th August, 2021

We appreciate the time and effort by the reviewer for reviewing this manuscript and for providing insightful comments and suggestions to improve its quality. We have revised the manuscript as per the suggestions and comments. The point-by-point responses are provided below.

Comments and Suggestions for Authors

General comment on the paper: In the manuscript “Distribution and molecular diversity of whitefly species colonizing cassava in Kenya and application of KASP genotyping to identify Bemisia tabaci haplotypes” the authors collect the samples of whiteflies across Kenya within several years and analyse the mitochondrial haplotypes. Due to challenging morphological species identification in whitefly, molecular methods are essential for this purpose. The authors carry out solid basic research in population genetics which is absolutely necessary for further pest management. The manuscript is well-written, easy to read and I suggest that is suitable for publication in the Insects journal. However, I have some suggestions and comments:

Response: Thank you for positive and encouraging general comments. We have incorporated the following specific comments in the revised version of manuscript. 

Specific comments

  1. I suggest shortening the title to its first part: “Distribution and molecular diversity of whitefly species colonizing cassava in Kenya” as KASP is used here as not the only method, it is not used for the first time and there is another publication by Wosula et al. 2020 which is dedicated to the application of KASP genotyping in Bemisia tabaci.

Response: Thank you and we agree with your suggestion. We have shorted the title to “Distribution and molecular diversity of whitefly species colonizing cassava in Kenya” and deleted the second part.

  1. For the collected samples, it would be good if the authors make an additional table in the supplementary to list all the samples with the locations, years, species, sample size, coordinates and other relevant information.

Response: Thank you for the observation. We have revised supplementary Table 1 and included the list of all samples with locations, years, species, sample size, coordinates and other information.

  1. I understand the information may be scarce but it would be good to add more discussion on species yearly distribution and population dynamics (if there is any knowledge on it) as you use samples of several years and they clearly differ across different years.

Response: Thank you for your comment. However, we did not get any information on species yearly distribution and population dynamics.  Please share any available manuscripts on the same.

  1. The authors could add a couple of sentences why they chose KASP genotyping over the other available methods

Response: KASP is a SNP based technique that is highly sensitive compared to other techniques. It was developed using sequence data generated through SNP genotyping (NextRAD) to characterize cassava colonizing Bemisia tabaci. SNP genotyping which is more robust compared to mtCO1 sequencing revealed six major haplogroups of cassava whiteflies. The mtCO1 was ineffective at distinguishing some of the haplogroups for example mitotype SSA-SG1 was found to have whiteflies that grouped into three distinct halopgroups SSA-ECA, SSA-CA and SSA-WA. In this study mitotypes SSA1-SG1 and SSA1-SG2 are all haplogroup SSA-ECA consistent to what has been reported in Wosula et al 2017, Chen et al 2019, Wosula et al 2020. KASP was developed to distinguish these SNP-based haplogroups under standard laboratory settings, and with end point fluorescent reads eliminates the cost and time delay incurred with mtCO1 sequencing. In this study both mtCO1 and KASP were used to highlight, KASP as a robust novel and rapid tool for identification of cassava B. tabaci.

KASP is a technique that was developed by LGC Genomics Teddington-UK, and it is based on allele-specific oligo extension and fluorescent resonance energy transfer (FRET) for signal generation. The assay uses allele-specific fluorescent labelled primers that allow for end-point fluorescent reads enabling the bi-allelic scoring of single nucleotide polymorphisms at specific loci (https://biosearch-cdn.azureedge.net/assetsv6/KASP-genotyping-chemistry-User-guide.pdf). KASP assays are low cost, high-throughput and have high specificity and sensitivity relative to other markers (Semagn et al 2014).

Additional suggestions:

Page 102 - The hyplotype names need the same punctuation: (SSA-1 to 5), SSA-2, SSA-3, SSA-4 and SSA-5.

Response: Thank you. We have used the same punctuations for the haplotypes as suggested. (Line 102 of the revised manuscript)

Page 144 - Global Positioning System - all capitals

Response: We have capitalized “p” in “Positioning” (Line 144). “Global Positioning System (GPS)” in the revised manuscript.

Figure2 – place pictures in alphabetical order

Response: Thank you. We have placed figure 2 pictures in alphabetical order in the revised manuscript

Figure3 is missing

Response: Thank you for the observation, we have noted and the anomaly has been corrected by revising the numbering of the figures.

Figure4 – make the same font                           

Response: Thank you. We noted the anomaly and changed the font to 9.

Reviewer 3 Report

Thr Manuscript ID insects-1335591 deals with the distribution and genetics of a crop pest, the whitefly, in Kenya. The MS is useful and can provide the journal with a good number of citations. However, the paper presents several flaws, and requires major revisions.

  1. Throughout the text, species name should be reported with genus and specific epithet ALWAYS in italics, with the genus in capital and the specific epithet in small letters. Please check throughout the MS starting from the title.
  2. Although I am not a native English speaker, I think that the MS requires a deep revision by a native speaker.
  3. Lines 65-66. USD 1 billion per year or per season?
  4. Lines 69-70. What do you mean by persistent and semipersistent?
  5. Lines 114-125. The end of the Introduction should include clear aims and numbered predictions, so to put the study in a more hypothesis-driven context.
  6. Methods are clear and replicable.
  7. You have “Figure 4” after “Figure 2”. You missed “Figure 3”.
  8. “Figure 4”, which should be “Figure 3”, includes several fonts. Please be consistent, and keep the species name in italics.
  9. “Figure 5”. Please use colour-blind, to allow everyone to appreciate differences.
  10. Line 451. Add a space between “species” and “ tabaci”.

Author Response

24th August, 2021

We appreciate the time and effort by the reviewer for reviewing this manuscript and for providing insightful comments and suggestions to improve its quality. We have revised the manuscript as per the suggestions and comments. The point-by-point responses are provided below.

Comments and Suggestions for Authors

General comment on the paper: The Manuscript ID insects-1335591 deals with the distribution and genetics of a crop pest, the whitefly, in Kenya. The MS is useful and can provide the journal with a good number of citations. However, the paper presents several flaws, and requires major revisions.

Response: We appreciate your time and valuable comments on the manuscript. We have incorporated the following specific comments in the revised version of manuscript. 

Specific comments

  1. Throughout the text, species name should be reported with genus and specific epithet ALWAYS in italics, with the genus in capital and the specific epithet in small letters. Please check throughout the MS starting from the title.

Response: Thank you; we have noted the anomaly throughout the manuscript. We have keenly checked through the whole manuscript and revised the genus name to start with a capital letter and species name with small letters.

  1. Although I am not a native English speaker, I think that the MS requires a deep revision by a native speaker.

Response: Thank you for your comment. We confirm that the manuscript has been read through and all the grammatical errors checked and corrected.

  1. Lines 65-66. USD 1 billion per year or per season?

Response: Thank you; we have corrected the anomaly. It is supposed to “USD 1 billion per year” as indicated in the revised manuscript.

  1. Lines 69-70. What do you mean by persistent and semi persistent?

Response:  Plant viruses can be transmitted by insect vectors in various ways depending on the length of the period the vector can harbor infectious particles. In this context, whitefly vectors which transmit both cassava mosaic and cassava brown streak viruses have been found to transmit in two different ways, persistently (vector can retain the virus for a longer period of time and efficiently transmit it) and  semi-persistently (vector can retain the virus for few hours) in transmission of cassava brown streak viruses. We have elaborated the terms in the manuscript.

  1. Lines 114-125. The end of the Introduction should include clear aims and numbered predictions, so to put the study in a more hypothesis-driven context.

Response: Thank you for the observation. We have included the following information at the end of the introduction. “There is inadequate information on the genetic variability and geographical distribution of whitefly species colonizing cassava in different cassava growing agro-ecological zones Kenya. Data are urgently required regarding the genetic groups, haplotype diversity, geographical distribution, and the phylogenetic relationships of whitefly species in Kenya. Therefore, the aim of the study was to determine the identity and distribution of the whitefly species as well as the genetic diversity of B. tabaci complex colonizing cassava”.

  1. Methods are clear and replicable.

Response: Thank you.

  1. You have “Figure 4” after “Figure 2”. You missed “Figure 3”.

Response: Thank you for the observation, we have noted and the anomaly has been corrected by revising the numbering of the figures.

  1. “Figure 4”, which should be “Figure 3”, includes several fonts. Please be consistent, and keep the species name in italics.

Response: Thank you. We noted the anomaly and changed the font to 10.

  1. Line 451. Add a space between “species” and “tabaci”.

Response: Thank you. We have revised the anomaly and added a space between “species” and “tabaci”.

Round 2

Reviewer 3 Report

Authors have now improved their MS which can now be accepted for publication on "Insects".

Author Response

Thank you.